The microbes we eat: abundance and taxonomy of microbes consumed in a day’s worth of meals for three diet types

Lang Jenna M. 1
Eisen Jonathan A. 2
Zivkovic Angela M. 3 4 amzivkovic@ucdavis.edu
1 Genome Center, University of California , Davis, CA , USA
2 Genome Center, Evolution and Ecology, Medical Microbiology and Immunology, University of California , Davis, CA , USA
3 Department of Nutrition, University of California , Davis, CA , USA
4 Foods for Health Institute, University of California , Davis, CA , USA
Kumar Abhishek
Electronic publication date: 2014 Dec 9
Publication date: 2014
Volume: 2
Electronic Location ID: e659
Received 2014 Aug 23; Accepted 2014 Oct 18
Copyright: © 2014 Lang et al.
Copyright year: 2014
Copyright holder: Lang et al.
License: This is an open access article distributed under the terms of the Creative Commons Attribution License, which permits unrestricted use, distribution, reproduction and adaptation in any medium and for any purpose provided that it is properly attributed. For attribution, the original author(s), title, publication source (PeerJ) and either DOI or URL of the article must be cited.
License URL: https://creativecommons.org/licenses/by/4.0/

Keywords: 16S, Microbial ecology, Microbiota, Microbiome, Bioinformatics, Microbial communities, Food microbiology, QIIME, PICRUSt, Illumina amplicon sequencing

Funding: Gordon and Betty Moore Foundation GBMF3330 University of California Discovery Program 09 GEB-02 NHB California Dairy Research Foundation This research is funded by the Gordon and Betty Moore Foundation through Grant GBMF3330 to Jonathan Eisen and the University of California Discovery Program 09 GEB-02 NHB, and the California Dairy Research Foundation. The funders had no role in study design, data collection and analysis, decision to publish, or preparation of the manuscript.

==============================
Far more attention has been paid to the microbes in our feces than the microbes in our food. Research efforts dedicated to the microbes that we eat have historically been focused on a fairly narrow range of species, namely those which cause disease and those which are thought to confer some “probiotic” health benefit. Little is known about the effects of ingested microbial communities that are present in typical American diets, and even the basic questions of which microbes, how many of them, and how much they vary from diet to diet and meal to meal, have not been answered.

We characterized the microbiota of three different dietary patterns in order to estimate: the average total amount of daily microbes ingested via food and beverages, and their composition in three daily meal plans representing three different dietary patterns. The three dietary patterns analyzed were: (1) the Average American (AMERICAN): focused on convenience foods, (2) USDA recommended (USDA): emphasizing fruits and vegetables, lean meat, dairy, and whole grains, and (3) Vegan (VEGAN): excluding all animal products. Meals were prepared in a home kitchen or purchased at restaurants and blended, followed by microbial analysis including aerobic, anaerobic, yeast and mold plate counts as well as 16S rRNA PCR survey analysis.

Based on plate counts, the USDA meal plan had the highest total amount of microbes at 1.3 × 109 CFU per day, followed by the VEGAN meal plan and the AMERICAN meal plan at 6 × 106 and 1.4 × 106 CFU per day respectively. There was no significant difference in diversity among the three dietary patterns. Individual meals clustered based on taxonomic composition independent of dietary pattern. For example, meals that were abundant in Lactic Acid Bacteria were from all three dietary patterns. Some taxonomic groups were correlated with the nutritional content of the meals. Predictive metagenome analysis using PICRUSt indicated differences in some functional KEGG categories across the three dietary patterns and for meals clustered based on whether they were raw or cooked.

Further studies are needed to determine the impact of ingested microbes on the intestinal microbiota, the extent of variation across foods, meals and diets, and the extent to which dietary microbes may impact human health. The answers to these questions will reveal whether dietary microbes, beyond probiotics taken as supplements—i.e., ingested with food—are important contributors to the composition, inter-individual variation, and function of our gut microbiota.

Introduction

The human gut microbiome (the total collection of microbes found in the human gut) mediates many key biological functions and its imbalance, termed dysbiosis, is associated with a number of inflammatory and metabolic diseases from inflammatory bowel disease to asthma to obesity and insulin resistance (Machonkin et al., 2014; Costello et al., 2012). How to effectively shift the microbiome and restore balance is a key question for disease prevention and treatment. The gut microbiome is influenced by a number of factors including the nature of the initial colonization at birth (e.g., vaginal vs. C-section delivery), host genotype, age, and diet. As diet is a readily modifiable factor, it is an obvious target for interventions. Several studies have confirmed high inter-individual variability in the bacterial composition of the gut microbiome in healthy individuals (Brownawell et al., 2012; Costello et al., 2009). Despite this high variability at the species level, enterotypes, or distinct clusters at the genus level, were described as core microbiomes that are independent of age, gender, nationality, or BMI (Arumugam et al., 2011). Although the concept of enterotypes is itself controversial, diet has been shown to play a key role in determining enterotype (Wu et al., 2011; De Filippo et al., 2010; Muegge et al., 2011). Although the core microbiota within each person are stable over longer time scales (e.g., 5 years), community composition is highly dynamic on shorter time scales (e.g., 0–50 weeks) (Faith et al., 2013). In fact, major shifts occur within 1 day of a significant dietary change (Wu et al., 2011; Turnbaugh et al., 2009). “Blooms” in specific bacterial groups were observed in response to controlled feeding of different fermentable fibers (Walker et al., 2011). Dietary changes affect both the structure and function of the gut microbiome in animals (Hildebrandt et al., 2009), and humans under controlled feeding conditions (Wu et al., 2011). Rapid shifts in microbiome composition are observed in response to change from a vegetarian to an animal based diet (David et al., 2013).

An ecological perspective helps to delineate the complexity and multi-layered nature of the relationships between the microbiota, the human host, and both the nutritive and non-nutritive compounds we ingest (Costello et al., 2012). The concept of the human gut microbiome as a distinct ecosystem or collection of micro-ecosystems allows us to identify and characterize the components of the system, including its inputs and outputs. In this case, the inputs of the system include all of the various ingested compounds that can either serve as food substrates (e.g., complex sugars) or that can be metabolized by or that affect the metabolism of the microbiota (e.g., polyphenolic compounds, environmental chemicals, medications). Some of these inputs, such as probiotics have been studied extensively. It has been well documented that certain sugars such as galactooligosaccharides, fructooligosaccharides, and oligosaccharides found in milk act as prebiotics that support the establishment and growth of certain commensal microbial species (Brownawell et al., 2012; de Vrese & Schrezenmeir, 2008; Roberfroid, 2007; German et al., 2008; Zivkovic & Barile, 2011). Research has also documented the effects of antibiotics, and pathogens on the microbiota composition, its recovery or lack of recovery to baseline following resolution, and the various immunological and physiological effects of these perturbations (Manichanh et al., 2010; Ubeda & Pamer, 2012; Bien, Palagani & Bozko, 2013; Dethlefsen & Relman, 2010).

Yet, little is known about the effects of ingested microorganisms on gut microbiota composition or function, and even the basic questions of which microbes, how many of them, and how much they vary from diet to diet and meal to meal, have not been answered. We do know about the microbial ecology of various specialty foods where fermentation, colonization, ripening, and/or aging are part of the preparation of these foods, for example pancetta (Busconi, Zacconi & Scolari, 2014) and of course cheese (Gatti et al., 2008; Button & Dutton, 2012). The microbial ecology of the surfaces of raw plant-derived foods such as fruits and vegetables has also been characterized (Leff & Fierer, 2013). There is a large base of literature on food-borne pathogens (Aboutaleb, Kuijper & Van Dissel, 2014). Furthermore, it is known that the microbial ecology of endemic microbes found on food surfaces can affect mechanisms by which pathogens colonize these foods (Critzer & Doyle, 2010). A recent article showed that certain ingested microbes found in foods such as cheese and deli meats were detected in the stool of individuals who consumed them, and that furthermore they were culturable and thus survived transit through the upper intestinal tract (David et al., 2013). However, the microbial ecology or microbial assemblages of different meals and diets, as well as the total number of live microorganisms ingested in these meals and diets are largely unknown. In fact, studies of the effects of diets and foods on the gut microbiota rely on dietary recalls and other dietary reporting instruments that were not designed to capture the potential variability in aspects of foods other than their basic macronutrient and micronutrient content. Specifically, current instruments for collecting individual dietary data do not capture the provenance of foods or their preparation, both of which would likely influence certain compositional aspects of the foods, especially the microbes on those foods.

We performed a preliminary study designed to generate hypotheses about the microbes we eat, and how they vary in terms of total abundance and relative composition in different meals and dietary patterns typical of American dietary intakes. We have selected to characterize the microbiota of 15 meals that exemplify the typical meals consumed as part of three different dietary patterns in order to determine the average total amount of daily microbes ingested via food and beverages and their composition in the average American adult consuming these typical foods/diets: (1) the Average American dietary pattern (AMERICAN) focused on convenience foods, (2) the USDA recommended dietary pattern (USDA) emphasizing fruits and vegetables, lean meat, dairy, and whole grains, and (3) the Vegan (VEGAN) dietary pattern, which excludes all animal products. We used DNA sequencing, plate counting, and informatics methods to characterize microbes in these meals and dietary patterns.

Methods

Meal preparation

We conducted a series of experiments consisting of food preparation followed by sample preparation and microbial analysis. Food was purchased and prepared in a standard American home kitchen by the same individual using typical kitchen cleaning practices including hand washing with non-antibacterial soap between food preparation steps, washing of dishes and cooking instruments with non-antibacterial dish washing detergent, and kitchen clean-up with a combination of anti-bacterial and non-antibacterial cleaning products. Anti-bacterial products had specific anti-bacterial molecules added to them whereas “non-antibacterial” products were simple surfactant-based formulations. The goal was to simulate a typical home kitchen rather than to artificially introduce sterile practices that would be atypical of how the average American prepares their meals at home. All meals were prepared according to specific recipes (from raw ingredient preparation such as washing and chopping, to cooking and mixing).

After food preparation, meals were plated on a clean plate, weighed on a digital scale (model 157W; Escali, Minneapolis, MN), and then transferred to a blender (model 5,200; Vita-Mix Corporation, Cleveland, OH) and processed until completely blended (approximately 1–3 min). Prepared, ready to eat foods that were purchased outside the home were simply weighed in their original packaging and then transferred to the blender. 4 mL aliquots of the blended meal composite were extracted from the blender, transported on dry ice and then stored at −80 °C until analysis. The following analyses were completed using these meal composite samples: (1) total aerobic bacterial plate counts, (2) total anaerobic bacterial plate counts, (3) yeast plate counts, (4) fungal plate counts, and (5) 16S rDNA analysis for microbial ecology.

Diet design

Diets were designed by a nutritional biologist to deliver the average number of calories consumed by an average American per day. The average American woman is 63 inches in height and weighs 166 pounds, and the average American man is 69 inches in height and weighs 195 pounds with an average age of 35, National Health and Nutrition Examination Survey, which translates to a total daily calorie intake range of 2,000–2,600 calories per day respectively to maintain weight, as determined using the USDA MyPlate SuperTracker tool. Therefore an intermediate daily calorie intake of about 2,200 calories was chosen as the target.

Meal plans were created using the NutriHand program (Nutrihand Inc., Soraya, CA). Diet nutrient composition was calculated by the NutriHand program from reference nutrient data for individual foods using the USDA National Nutrient Database for Standard Reference. Three one-day meal plans were created to be representative of three typical dietary patterns that are consumed by Americans: (1) the Average American dietary pattern (AMERICAN), which includes meat and dairy and focuses on convenience foods, (2) the USDA recommended dietary pattern (USDA), which emphasizes fresh fruits and vegetables, lean meats, whole grains and whole grain products, and dairy, and (3) the Vegan dietary pattern (VEGAN), which excludes all animal products. The AMERICAN meal plan totaled 2,268 calories, which consisted of 35% fat, 53% carbohydrates of which 16.6 g was fiber, and 12% protein. The USDA meal plan totaled 2,260 calories, consisting of 25% fat, 49% carbohydrates of which 45 g was fiber, and 27% protein. The VEGAN meal plan totaled 2,264 calories and consisted of 31% fat, 54% carbohydrates of which 52 g was fiber, and 15% protein.

Microbial community analysis

Microbial plate counts were performed by Covance Laboratories (Covance Inc., Madison, WI). Aerobic plate counts were performed according to SPCM:7, anaerobic plate counts were performed according to APCM:5 and the yeast and mold counts were performed according to Chapter 23 of the FDA’s Bacteriological Analytical Manual. Plate counts were reported as colony forming units (CFU) per gram for each meal composite. The CFU/g values were multiplied by the total number of grams in each meal to obtain the CFU per meal, and the values for meals for each day were added to obtain the CFU per day for each dietary pattern (Table 1).

Table 1 Aerobic and anaerobic microbial plate counts.

Bacterial (aerobic and anaerobic), yeast, and mold plate counts were performed by Covance Laboratories (Covance Inc., Madison, WI). Plate counts are reported as colony forming units (CFU) per gram for each meal.

Dietary pattern	Meal	Aerobic plate
count	Anaerobic plate
count	Yeast count	Mold count	Total microorganisms	
Average American	Breakfast	2.15E+05	2.26E+05	5.66E+02	5.66E+03	4.48E+05	
	Lunch	2.23E+05	1.31E+04	1.31E+03	1.31E+03	2.38E+05	
	Snack	1.87E+04	2.34E+03	2.34E+02	2.34E+02	2.15E+04	
	Dinner	1.47E+05	5.35E+05	7.75E+02	7.75E+02	6.84E+05	
	Total	6.04E+05	7.77E+05	2.88E+03	7.98E+03	1.39E+06	
USDA recommended	Breakfast	1.14E+04	5.72E+02	4.29E+04	1.49E+06	1.54E+06	
	Snack #1	2.11E+05	5.42E+07	5.42E+02	1.19E+05	5.45E+07	
	Lunch	3.25E+07	2.26E+06	1.06E+06	2.55E+06	3.84E+07	
	Snack #2	5.54E+08	6.09E+08	3.32E+04	1.39E+04	1.16E+09	
	Dinner	3.49E+05	5.81E+04	9.69E+02	9.69E+03	4.17E+05	
	Total	5.87E+08	6.66E+08	1.14E+06	4.18E+06	1.26E+09	
Vegan	Breakfast	3.38E+04	1.99E+04	3.98E+02	9.95E+03	6.41E+04	
	Snack #1	1.97E+06	1.67E+06	5.41E+05	4.92E+05	4.67E+06	
	Lunch	1.22E+05	2.34E+04	9.35E+02	9.35E+02	1.47E+05	
	Snack #2	9.81E+04	4.67E+03	3.50E+04	2.80E+05	4.18E+05	
	Dinner	4.07E+05	1.45E+05	4.53E+02	9.05E+03	5.62E+05	
	Snack #3	1.43E+05	2.94E+03	8.40E+01	8.40E+01	1.46E+05	
	Total	2.77E+06	1.87E+06	5.78E+05	7.92E+05	6.01E+06	

The taxonomic composition of each meal microbiome was assessed via amplification and sequencing of 16S rDNA from the homogenized meals. DNA was extracted from homogenized food samples with the Power Food Microbial DNA Isolation Kit (MoBio Laboratories, Inc.) according to the manufacturer’s protocol. Microbial DNA was amplified by a two-step PCR enrichment of the 16S rRNA gene (V4 region) using primers 515F and 806R, modified by addition of Illumina adaptor and barcodes sequences. All primer sequences and a detailed PCR protocol are provided in Table 2 and in a GitHub repository (https://github.com/hollybik/protocols/blob/master/16S_rRNA_twostep_PCR.tex), respectively. Libraries were sequenced using an Illumina MiSeq system, generating 250bp paired-end amplicon reads. The amplicon data was multiplexed using dual barcode combinations for each sample. We used a custom script (available in a GitHub repository (https://github.com/gjospin/scripts/blob/master/Demul_trim_prep.pl), to assign each pair of reads to their respective samples when parsing the raw data. This script allows for 1 base pair difference per barcode. The paired reads were then aligned and a consensus was computed using FLASH (Magoč & Salzberg, 2011) with maximum overlap of 120 and a minimum overlap of 70 (other parameters were left as default). The custom script automatically demultiplexes the data into fastq files, executes FLASH, and parses its results to reformat the sequences with appropriate naming conventions for QIIME v.1.8.0 (Caporaso et al., 2010) in fasta format. The resulting consensus sequences were analyzed using the QIIME pipeline.

Table 2 PCR primer constructs for Illumina MiSeq 16S rDNA amplicon sequencing.

16S rDNA PCR was performed using these primers that include adaptors necessary for binding to the Illumina MiSeq flow cell, a spacer sequence, an 8bp barcode sequence, and the “universal” 16S rDNA primers 515F and 806R.

Name	Illumina adapter sequence	Barcode	Pad	Linker	Primer	Complete oligo	
16S515bc1	AATGATACGGCGACC
ACCGAGATCTACAC	AACCAGTC	TATGGTAATTG	TG	TGCCAGCMGCCGCGGTAA	AATGATACGGCGACC
ACCGAGATCTACACAA
CCAGTCTATGGTAA
TTGTGTGCCAGC
MGCCGCGGTAA	
16S515bc2	AATGATACGGCGACC
ACCGAGATCTACAC	AACGCTAA	TATGGTAATTG	TG	TGCCAGCMGCCGCGGTAA	AATGATACGGCGACC
ACCGAGATCTACACAA
CGCTAATATGGTAA
TTGTGTGCCAGC
MGCCGCGGTAA	
16S515bc3	AATGATACGGCGACC
ACCGAGATCTACAC	AAGACTAC	TATGGTAATTG	TG	TGCCAGCMGCCGCGGTAA	AATGATACGGCGACC
ACCGAGATCTACACAA
GACTACTATGGTAA
TTGTGTGCCAGC
MGCCGCGGTAA	
16S515bc4	AATGATACGGCGACC
ACCGAGATCTACAC	AATCGATA	TATGGTAATTG	TG	TGCCAGCMGCCGCGGTAA	AATGATACGGCGACC
ACCGAGATCTACACAA
TCGATATATGGTAA
TTGTGTGCCAGC
MGCCGCGGTAA	
16S515bc5	AATGATACGGCGACC
ACCGAGATCTACAC	ACCAATTG	TATGGTAATTG	TG	TGCCAGCMGCCGCGGTAA	AATGATACGGCGACC
ACCGAGATCTACACACC
AATTGTATGGTAA
TTGTGTGCCAGC
MGCCGCGGTAA	
16S515bc6	AATGATACGGCGACC
ACCGAGATCTACAC	ACTGAAGT	TATGGTAATTG	TG	TGCCAGCMGCCGCGGTAA	AATGATACGGCGACC
ACCGAGATCTACACACT
GAAGTTATGGTAA
TTGTGTGCCAGC
MGCCGCGGTAA	
16S515bc7	AATGATACGGCGACC
ACCGAGATCTACAC	ATTGCCGC	TATGGTAATTG	TG	TGCCAGCMGCCGCGGTAA	AATGATACGGCGACC
ACCGAGATCTACACATT
GCCGCTATGGTAA
TTGTGTGCCAGC
MGCCGCGGTAA	
16S515bc8	AATGATACGGCGACC
ACCGAGATCTACAC	CAACCTTA	TATGGTAATTG	TG	TGCCAGCMGCCGCGGTAA	AATGATACGGCGACC
ACCGAGATCTACACCAA
CCTTATATGGTAA
TTGTGTGCCAGC
MGCCGCGGTAA	
16S515bc9	AATGATACGGCGACC
ACCGAGATCTACAC	CCTAATAA	TATGGTAATTG	TG	TGCCAGCMGCCGCGGTAA	AATGATACGGCGACC
ACCGAGATCTACACCCT
AATAATATGGTAA
TTGTGTGCCAGC
MGCCGCGGTAA	
16S515bc10	AATGATACGGCGACC
ACCGAGATCTACAC	CCTCTGAT	TATGGTAATTG	TG	TGCCAGCMGCCGCGGTAA	AATGATACGGCGACC
ACCGAGATCTACACCC
TCTGATTATGGTAA
TTGTGTGCCAGC
MGCCGCGGTAA	
16S515bc11	AATGATACGGCGACC
ACCGAGATCTACAC	CGGTCGAG	TATGGTAATTG	TG	TGCCAGCMGCCGCGGTAA	AATGATACGGCGACC
ACCGAGATCTACACCG
GTCGAGTATGGTAA
TTGTGTGCCAGC
MGCCGCGGTAA	
16S515bc12	AATGATACGGCGACC
ACCGAGATCTACAC	CTAATGGC	TATGGTAATTG	TG	TGCCAGCMGCCGCGGTAA	AATGATACGGCGACC
ACCGAGATCTACACCT
AATGGCTATGGTAA
TTGTGTGCCAGC
MGCCGCGGTAA	
16S515bc13	AATGATACGGCGACC
ACCGAGATCTACAC	CTCATGCG	TATGGTAATTG	TG	TGCCAGCMGCCGCGGTAA	AATGATACGGCGACC
ACCGAGATCTACACCT
CATGCGTATGGTAA
TTGTGTGCCAGC
MGCCGCGGTAA	
16S515bc14	AATGATACGGCGACC
ACCGAGATCTACAC	GAACGGAG	TATGGTAATTG	TG	TGCCAGCMGCCGCGGTAA	AATGATACGGCGACC
ACCGAGATCTACACGA
ACGGAGTATGGTAA
TTGTGTGCCAGC
MGCCGCGGTAA	
16S515bc15	AATGATACGGCGACC
ACCGAGATCTACAC	GCCTACGC	TATGGTAATTG	TG	TGCCAGCMGCCGCGGTAA	AATGATACGGCGACC
ACCGAGATCTACACGC
CTACGCTATGGTAA
TTGTGTGCCAGC
MGCCGCGGTAA	
16S515bc16	AATGATACGGCGACC
ACCGAGATCTACAC	GCGTTACC	TATGGTAATTG	TG	TGCCAGCMGCCGCGGTAA	AATGATACGGCGACC
ACCGAGATCTACACGC
GTTACCTATGGTAA
TTGTGTGCCAGC
MGCCGCGGTAA	
16S515bc17	AATGATACGGCGACC
ACCGAGATCTACAC	GGAGGCTG	TATGGTAATTG	TG	TGCCAGCMGCCGCGGTAA	AATGATACGGCGACC
ACCGAGATCTACACGG
AGGCTGTATGGTAA
TTGTGTGCCAGC
MGCCGCGGTAA	
16S515bc18	AATGATACGGCGACC
ACCGAGATCTACAC	GGATGCCA	TATGGTAATTG	TG	TGCCAGCMGCCGCGGTAA	AATGATACGGCGACC
ACCGAGATCTACACGG
ATGCCATATGGTAA
TTGTGTGCCAGC
MGCCGCGGTAA	
16S515bc19	AATGATACGGCGACC
ACCGAGATCTACAC	GGATTAGG	TATGGTAATTG	TG	TGCCAGCMGCCGCGGTAA	AATGATACGGCGACC
ACCGAGATCTACACGG
ATTAGGTATGGTAA
TTGTGTGCCAGC
MGCCGCGGTAA	
16S515bc20	AATGATACGGCGACC
ACCGAGATCTACAC	GTTGGCCG	TATGGTAATTG	TG	TGCCAGCMGCCGCGGTAA	AATGATACGGCGACC
ACCGAGATCTACACGT
TGGCCGTATGGTAA
TTGTGTGCCAGC
MGCCGCGGTAA	
16S515bc21	AATGATACGGCGACC
ACCGAGATCTACAC	TATTAACT	TATGGTAATTG	TG	TGCCAGCMGCCGCGGTAA	AATGATACGGCGACC
ACCGAGATCTACACTA
TTAACTTATGGTAA
TTGTGTGCCAGC
MGCCGCGGTAA	
16S515bc22	AATGATACGGCGACC
ACCGAGATCTACAC	TGACTGCT	TATGGTAATTG	TG	TGCCAGCMGCCGCGGTAA	AATGATACGGCGACC
ACCGAGATCTACACTG
ACTGCTTATGGTAA
TTGTGTGCCAGC
MGCCGCGGTAA	
16S515bc23	AATGATACGGCGACC
ACCGAGATCTACAC	TGGCGATT	TATGGTAATTG	TG	TGCCAGCMGCCGCGGTAA	AATGATACGGCGACC
ACCGAGATCTACACTG
GCGATTTATGGTAA
TTGTGTGCCAGC
MGCCGCGGTAA	
16S515bc24	AATGATACGGCGACC
ACCGAGATCTACAC	TTCAGCGA	TATGGTAATTG	TG	TGCCAGCMGCCGCGGTAA	AATGATACGGCGACC
ACCGAGATCTACACTT
CAGCGATATGGTAA
TTGTGTGCCAGC
MGCCGCGGTAA	
16S515bc25	AATGATACGGCGACC
ACCGAGATCTACAC	TTGGCTAT	TATGGTAATTG	TG	TGCCAGCMGCCGCGGTAA	AATGATACGGCGACC
ACCGAGATCTACACTT
GGCTATTATGGTAA
TTGTGTGCCAGC
MGCCGCGGTAA	
16S806Rbc1	AATGATACGGCGACC
ACCGAGATCTACAC	AACCAGTC	AGTCAGTCAG	CC	GGACTACHVGGGTWTCTAAT	AATGATACGGCGACC
ACCGAGATCTACACAA
CCAGTCAGTCAGTC
AGCCGGACTACH
VGGGTWTCTAAT	
16S806Rbc2	AATGATACGGCGACC
ACCGAGATCTACAC	AACGCTAA	AGTCAGTCAG	CC	GGACTACHVGGGTWTCTAAT	AATGATACGGCGACC
ACCGAGATCTACACAA
CGCTAAAGTCAGTC
AGCCGGACTACH
VGGGTWTCTAAT	
16S806Rbc3	AATGATACGGCGACC
ACCGAGATCTACAC	AAGACTAC	AGTCAGTCAG	CC	GGACTACHVGGGTWTCTAAT	AATGATACGGCGACC
ACCGAGATCTACACAA
GACTACAGTCAGTC
AGCCGGACTACH
VGGGTWTCTAAT	
16S806Rbc4	AATGATACGGCGACC
ACCGAGATCTACAC	AATCGATA	AGTCAGTCAG	CC	GGACTACHVGGGTWTCTAAT	AATGATACGGCGACC
ACCGAGATCTACACAA
TCGATAAGTCAGTC
AGCCGGACTACH
VGGGTWTCTAAT	
16S806Rbc5	AATGATACGGCGACC
ACCGAGATCTACAC	ACCAATTG	AGTCAGTCAG	CC	GGACTACHVGGGTWTCTAAT	AATGATACGGCGACC
ACCGAGATCTACACAC
CAATTGAGTCAGTC
AGCCGGACTACH
VGGGTWTCTAAT	
16S806Rbc6	AATGATACGGCGACC
ACCGAGATCTACAC	ACTGAAGT	AGTCAGTCAG	CC	GGACTACHVGGGTWTCTAAT	AATGATACGGCGACC
ACCGAGATCTACACACT
GAAGTAGTCAGTC
AGCCGGACTACH
VGGGTWTCTAAT	
16S806Rbc7	AATGATACGGCGACC
ACCGAGATCTACAC	ATTGCCGC	AGTCAGTCAG	CC	GGACTACHVGGGTWTCTAAT	AATGATACGGCGACC
ACCGAGATCTACACATT
GCCGCAGTCAGTC
AGCCGGACTACH
VGGGTWTCTAAT	
16S806Rbc8	AATGATACGGCGACC
ACCGAGATCTACAC	CAACCTTA	AGTCAGTCAG	CC	GGACTACHVGGGTWTCTAAT	AATGATACGGCGACC
ACCGAGATCTACACCAA
CCTTAAGTCAGTC
AGCCGGACTACH
VGGGTWTCTAAT	
16S806Rbc9	AATGATACGGCGACC
ACCGAGATCTACAC	CCTAATAA	AGTCAGTCAG	CC	GGACTACHVGGGTWTCTAAT	AATGATACGGCGACC
ACCGAGATCTACACCCT
AATAAAGTCAGTC
AGCCGGACTACH
VGGGTWTCTAAT	
16S806Rbc10	AATGATACGGCGACC
ACCGAGATCTACAC	CCTCTGAT	AGTCAGTCAG	CC	GGACTACHVGGGTWTCTAAT	AATGATACGGCGACC
ACCGAGATCTACACCC
TCTGATAGTCAGTC
AGCCGGACTACH
VGGGTWTCTAAT	
16S806Rbc11	AATGATACGGCGACC
ACCGAGATCTACAC	CGGTCGAG	AGTCAGTCAG	CC	GGACTACHVGGGTWTCTAAT	AATGATACGGCGACC
ACCGAGATCTACACCG
GTCGAGAGTCAGTC
AGCCGGACTACH
VGGGTWTCTAAT	
16S806Rbc12	AATGATACGGCGACC
ACCGAGATCTACAC	CTAATGGC	AGTCAGTCAG	CC	GGACTACHVGGGTWTCTAAT	AATGATACGGCGACC
ACCGAGATCTACACCT
AATGGCAGTCAGTC
AGCCGGACTACH
VGGGTWTCTAAT	
16S806Rbc13	AATGATACGGCGACC
ACCGAGATCTACAC	CTCATGCG	AGTCAGTCAG	CC	GGACTACHVGGGTWTCTAAT	AATGATACGGCGACC
ACCGAGATCTACACCT
CATGCGAGTCAGTC
AGCCGGACTACH
VGGGTWTCTAAT	
16S806Rbc14	AATGATACGGCGACC
ACCGAGATCTACAC	GAACGGAG	AGTCAGTCAG	CC	GGACTACHVGGGTWTCTAAT	AATGATACGGCGACC
ACCGAGATCTACACGA
ACGGAGAGTCAGTC
AGCCGGACTACH
VGGGTWTCTAAT	
16S806Rbc15	AATGATACGGCGACC
ACCGAGATCTACAC	GCCTACGC	AGTCAGTCAG	CC	GGACTACHVGGGTWTCTAAT	AATGATACGGCGACC
ACCGAGATCTACACGC
CTACGCAGTCAGTC
AGCCGGACTACH
VGGGTWTCTAAT	
16S806Rbc16	AATGATACGGCGACC
ACCGAGATCTACAC	GCGTTACC	AGTCAGTCAG	CC	GGACTACHVGGGTWTCTAAT	AATGATACGGCGACC
ACCGAGATCTACACGC
GTTACCAGTCAGTC
AGCCGGACTACH
VGGGTWTCTAAT	
16S806Rbc17	AATGATACGGCGACC
ACCGAGATCTACAC	GGAGGCTG	AGTCAGTCAG	CC	GGACTACHVGGGTWTCTAAT	AATGATACGGCGACC
ACCGAGATCTACACGG
AGGCTGAGTCAGTC
AGCCGGACTACH
VGGGTWTCTAAT	
16S806Rbc18	AATGATACGGCGACC
ACCGAGATCTACAC	GGATGCCA	AGTCAGTCAG	CC	GGACTACHVGGGTWTCTAAT	AATGATACGGCGACC
ACCGAGATCTACACGG
ATGCCAAGTCAGTC
AGCCGGACTACH
VGGGTWTCTAAT	
16S806Rbc19	AATGATACGGCGACC
ACCGAGATCTACAC	GGATTAGG	AGTCAGTCAG	CC	GGACTACHVGGGTWTCTAAT	AATGATACGGCGACC
ACCGAGATCTACACGG
ATTAGGAGTCAGTC
AGCCGGACTACH
VGGGTWTCTAAT	
16S806Rbc20	AATGATACGGCGACC
ACCGAGATCTACAC	GTTGGCCG	AGTCAGTCAG	CC	GGACTACHVGGGTWTCTAAT	AATGATACGGCGACC
ACCGAGATCTACACGT
TGGCCGAGTCAGTC
AGCCGGACTACH
VGGGTWTCTAAT	
16S806Rbc21	AATGATACGGCGACC
ACCGAGATCTACAC	TATTAACT	AGTCAGTCAG	CC	GGACTACHVGGGTWTCTAAT	AATGATACGGCGACC
ACCGAGATCTACACTA
TTAACTAGTCAGTC
AGCCGGACTACH
VGGGTWTCTAAT	
16S806Rbc22	AATGATACGGCGACC
ACCGAGATCTACAC	TGACTGCT	AGTCAGTCAG	CC	GGACTACHVGGGTWTCTAAT	AATGATACGGCGACC
ACCGAGATCTACACTG
ACTGCTAGTCAGTC
AGCCGGACTACH
VGGGTWTCTAAT	
16S806Rbc23	AATGATACGGCGACC
ACCGAGATCTACAC	TGGCGATT	AGTCAGTCAG	CC	GGACTACHVGGGTWTCTAAT	AATGATACGGCGACC
ACCGAGATCTACACTG
GCGATTAGTCAGTC
AGCCGGACTACH
VGGGTWTCTAAT	
16S806Rbc24	AATGATACGGCGACC
ACCGAGATCTACAC	TTCAGCGA	AGTCAGTCAG	CC	GGACTACHVGGGTWTCTAAT	AATGATACGGCGACC
ACCGAGATCTACACTT
CAGCGAAGTCAGTC
AGCCGGACTACH
VGGGTWTCTAAT	
16S806Rbc25	AATGATACGGCGACC
ACCGAGATCTACAC	TTGGCTAT	AGTCAGTCAG	CC	GGACTACHVGGGTWTCTAAT	AATGATACGGCGACC
ACCGAGATCTACACTT
GGCTATAGTCAGTC
AGCCGGACTACH
VGGGTWTCTAAT	

Statistical analyses and data visualization

Unless otherwise noted, all statistical analyses were performed using python scripts implemented in QIIME v.1.8.0, and all python scripts referenced here are QIIME scripts. The IPython notebook file used for all QIIME analyses is available at http://nbviewer.ipython.org/gist/jennomics/c6fe5e113525c6aa8add. To explore the differences in overall microbial community composition across the 15 meals, both the phylogenetic weighted UniFrac distances (Lozupone et al., 2011) and the taxonomic Bray–Curtis dissimilarities (Bray & Curtis, 1957) were calculated using the beta_diversity_through_plots.py script. This script also produced a principal coordinates analysis (PCoA) plot in which the Bray–Curtis dissimilarities between samples were used to visualize differences among groups of samples (see Fig. 1 for this type of visualization for the three Diet Types.) To test for the significance of dietary pattern on the overall microbial community composition, we used a permutational multivariate ANOVA as implemented in the compare_categories.py script. To test for significant differences in taxonomic richness across dietary patterns, we used the non-parametric Kruskal–Wallis test (Kruskal & Allen Wallis, 1952) with the FDR (false discovery rate) correction as implemented in compare_alpha_diversity.py. To test for the significant variation in frequency of individual OTUs across dietary patterns, we used the Kruskal–Wallis test with the FDR correction as implemented in the group_significance.py script. We also used the biplot function of the make_emperor.py script to plot the family-level OTUs in PCoA space alongside each meal. To test for significant correlation between the relative abundance of a single taxonomic group and meal metadata categories (i.e., nutrient composition, whether a meal contains fermented foods, etc.) at 5 taxonomic levels (phylum-genus) Pearson correlation coefficients (Pearson, 1895) were calculated and tested for statistical significance using Stata (Stata Statistical Software Release 13; StataCorp, College Station, TX). Figures 2 and 3 were produced with R (R-project, 2014), using the phyloseq package (McMurdie & Holmes, 2013).

Figure 1 Principle Coordinates Analysis plot.

Principle Coordinates Analysis (PCoA) based on Bray–Curtis dissimilarities of microbial communities found in the 15 meals, color-coded according to the dietary patterns they represent. Axes are scaled to the amount of variation explained.

Metagenome prediction with PICRUSt

A synthetic metagenome was generated based on the observed 16S rDNA sequences for each meal. To do this, the 16S rDNA sequences were clustered into a collection of OTUs sharing 99% sequence identity, using the pick_closed_reference_otus.py script. The resultant OTU table was normalized with respect to inferred 16S rRNA gene copy numbers using the normalize_by_copy_number.py script distributed with PICRUSt v.1.0.0 (Langille et al., 2013). The normalized OTU table was used to predict meal microbial metagenomes with PICRUSt’s predict_metagenomes.py script. The final predicted metagenome is output as a .biom table, which is suitable for analysis with a tool such as STAMP (Parks et al., 2014). We used STAMP to test for and visualize significant (predicted) functional differences in microbial communities between the three dietary patterns.

Results

Meal composition

The detailed meal plans with all ingredients are shown in Table 3, food preparation descriptions (all steps prior to placing into blender and blending foods as described in “Methods” section) are shown in Table 4, and nutrient values based on the USDA nutrient database are shown in Table 5. The AMERICAN meal plan consisted of a large Starbucks Mocha Frapuccino for breakfast, a McDonald’s Big Mac, French fries, and Coca Cola for lunch, Stouffer’s lasagna for dinner, and Oreo cookies for a snack. The USDA meal plan consisted of cereal with milk and raspberries for breakfast, an apple and yogurt for a morning snack, a turkey sandwich on whole wheat bread with salad (including a hard-boiled egg, grapes, parmesan cheese, and Ceasar dressing) for lunch, carrots, cottage cheese and chocolate chips for an afternoon snack, and chicken, asparagus, peas and spinach on quinoa for dinner. The VEGAN meal plan consisted of oatmeal with banana, peanut butter, and almond milk for breakfast, a protein shake (including vegetable-based protein powder, soy milk, banana and blueberries) for a morning snack, a vegetable and tofu soup (including soba noodles, spinach, carrots, celery and onions in vegetable broth) for lunch, an apple and almonds with tea for an afternoon snack, a Portobello mushroom burger (including Portobello mushroom, avocado, tomato, lettuce, and a whole wheat bun) with steamed broccoli for dinner, and popcorn, hazelnuts and fig bars for an evening snack.

Table 3 Ingredients included in each meal.

A detailed accounting of each component of each meal, including the weight of each ingredient.

	Average American	USDA	Vegan	
	Amount	Item	Amount	Item	Amount	Item	
Breakfast	20 oz (566 g)	Starbucks
Mocha Frappucino	2 cups (88 g)	Kashi
GoLean cereal	0.5 each (60 g)	large banana	
			1 cup (232 g)	1% milk	1 cup (250 g)	Almond Breeze
almond milk	
			0.5 cup (58 g)	raspberries	2 tsp (14 g)	maple syrup	
					1 tbsp (28 g)	peanut butter	
					0.5 cup (46 g)	rolled oats	
Lunch	1 each (215 g)	McDonald’s Big Mac	2 tbsp (26 g)	Cesar dressing	6 oz (171 g)	firm tofu	
	1 large (154 g)	McDonald’s French Fries	20 each (125 g)	green seedless grapes	2 oz (57 g)	soba noodles	
	12 fl oz (380 g)	McDonald’s Coke	1 each (78 g)	Oroweat whole wheat
burger bun	1 cup (28 g)	spinach	
			3 cups (72 g)	green leaf lettuce	1 each (71 g)	medium carrot	
			1 each (52 g)	large hard boiled egg	2 cups (480 g)	Pacific Foods
vegetable broth	
			3 tbsp (18 g)	parmesan cheese, shredded	1 stalk (56 g)	medium celery	
			2 slices (46 g)	roasted turkey breast	0.25 cup (65 g)	chopped yellow onion	
					1 tsp (5 g)	extra virgin olive oil	
					0.25 tsp (2 g)	toasted sesame oil	
Dinner	2 slices (515 g)	Stouffer’s Lasagna	1 tbsp (12 g)	extra virgin olive oil	0.25 each (38 g)	avocado	
			1 cup (171 g)	quinoa	1 each (159 g)	portabella mushroom	
			0.33 cup (35 g)	diced yellow onion	1 tbsp (14 g)	balsamic vinegar	
			4 each (65 g)	medium asparagus spears	1 tbsp (14 g)	Vegenaise	
			0.5 cup (72 g)	frozen green peas	1 slice (57 g)	tomato	
			6 oz (165 g)	boneless skinless
chicken poached	1 leaf (14 g)	red lettuce	
			0.5 cup (13 g)	spinach	1 cup (80 g)	chopped broccoli	
			1 tsp (0.5 g)	lemon juice	1 tsp (0.5 g)	lemon juice	
			1.5 cup (435 g)	water	1 clove (0.5 g)	garlic	
					1 tbsp	chopped basil	
					1 bun	Oroweat whole wheat
burger bun	
Snack #1			1 each	small Fuji apple	0.5 each	large banana	
			6 oz	Yoplait strawberry yogurt	1 cup	soy milk	
					1 scoop	Spirutein protein powder	
					1 cup	blueberries (Chile)	
Snack #2	3 each	Oreo cookies	10 each	large baby carrots	1 bag	green tea	
			1 cup	2% cottage cheese	1 cup	water	
			2 tbsp	semi-sweet chocolate chips	1 each	medium Fuji apple	
					20 each	almonds	
Snack #3					2 cups	pop corn	
					17 each	hazelnuts	
					3 each	Newman’s Own fig bars	

Table 4 Food preparation details.

The meal preparation and cooking instructions (when appropriate) are presented here.

	Average American	USDA	Vegan	
Breakfast	Used as purchased from Starbucks.	Cereal poured directly from box into bowl. Milk poured into measuring cup, then into cereal bowl. Raspberries washed first in colander under running water then transferred on top of cereal.	Almond milk brought to a boil, then oats added and cooked for 5 min on low heat. Peanut butter and maple syrup measured out then stirred into cooked oats. Banana peeled and sliced into slices on top of cooked oats.	
Snack #1	Cookies taken out of packaging.	Apple washed and sliced, core discarded. Yogurt used as purchased.	Soy milk measured into measuring cup, protein powder measured into scoop, banana peeled and cut in half, blueberries rinsed in colander under running water.	
Lunch	Used as purchased from McDonald’s.	Sliced roasted turkey breast deli meat taken out of packaging and placed into burger bun. Lettuce rinsed in colander under running water and dried on paper towel then cut into strips and tossed with premade Ceasar dressing. Egg boiled in water for 8 min then peeled and sliced in half and placed in top of dressed salad. Parmesan cheese shredded and added on top of salad. Grapes rinsed in colander under running water, then sliced in half and placed on top of salad.	Carrot rinsed under running water, peeled, and sliced. Celery washed under running water and sliced. Onion outer layer peeled and diced. Sliced carrot, celery and onion sauteed in olive oil for 5 min, then vegetable broth measured out in measuring cup and added to vegetables, brought to a boil. Tofu taken out of packaging, excess water discarded, cut into cubes, added to broth. Spinach taken out of prepackaged, prerinsed bag and added to broth. Noodles and sesame oil added to broth. Soup cooked for 8 min on low heat.	
Snack #2		Baby carrots taken out of packaging and used. Cottage cheese measured out in measuring cup. Chocolate chips measured out and used.	Water boiled and poured into cup with tea bag, steeped for 5 min. Apple rinsed under running water, sliced, and core discarded. Almonds taken out of packaging.	
Dinner	Lasagna prepared according to manufacturer instructions (taken out of freezer and baked at 400F for 1 h and 45 min, cooled, then sliced.	Chicken breast taken out of plastic packaging, and placed into pot with boiling water, boiled for 3 min, removed from heat, covered, let stand for 18 min, then sliced. Quinoa rinsed in colander under running water, added to water in pan and brought to a boil, simmered covered for 20 min. Oil heated in large skillet over medium heat, onion peeled and diced, asparagus spears rinsed and sliced, both added to oil and cooked for 5 min. Peas added from frozen packaging and cooked for 1 min. Spinach rinsed in colander under running water and added to skillet, cooked for 3 min. Quinoa, vegetables, and chicken combined with lemon juice.	Mushroom destemmed and peeled, soaked in vinegar, then grilled in grill pan for 5 min on each side. Garlic peeled and grated into Vegenaise, lemon juice added. Basil rinsed under running water, chopped and added to Vegenaise mixture. Tomato rinsed under running water, then sliced. Lettuce leaf rinsed under running water. Broccoli rinsed under running water, then steamed in colander for 3 min, chopped. Burger assembled: Vegenaise mixture spread onto bottom of bun, topped with mushroom, lettuce leaf, tomato slice and top of bun.	
Snack #3			Popcorn (no salt, no oil) prepared in microwave bag as directed (placed in microwave for 4 min). Hazelnuts taken out of packaging. Fig bars taken out of packaging.	

Table 5 Nutrient composition by meal.

Diet nutrient composition was calculated by the NutriHand program from reference nutrient data for individual foods using the USDA National Nutrient Database for Standard Reference.

Dietary
pattern/meal	Energy
(kcal)	Protein
(g)	Total lipid
(fat) (g)	Carbohydrate,
by difference
(g)	Fiber, total
dietary (g)	Sugars,
total (g)	Calcium,
Ca (mg)	Iron,
Fe (mg)	
AMERICAN breakfast	367	8.52	4.98	73.33	0	60	250	1.2	
AMERICAN lunch	1,174	31.76	57.58	138.6	10	43.53	280	5.7	
AMERICAN dinner	568	26.62	20.81	68.2	5.6	11.4	380	2.88	
AMERICAN snack	160	1	7	25	1	14	20	1.8	
USDA breakfast	414	34.96	4.85	79.52	24	27.41	466	4.09	
USDA snack #1	256	8.91	3.07	52.29	3.6	20.45	268	0.3	
USDA lunch	656	37.26	28.97	68.61	6	19.06	298	6.06	
USDA snack #2	352	29.2	11.74	34.13	5.8	27.43	254	2.22	
USDA dinner	581	48.44	19.22	56.41	5.6	6.97	58	3	
VEGAN breakfast	367	10	12.78	58.19	6.7	28.31	311	0.9	
VEGAN snack #1	373	23.76	4.95	62.8	7.7	40.46	373	6.65	
VEGAN lunch	468	25.49	13.89	64.31	7	7.95	348	6.94	
VEGAN snack #2	233	5.82	12.59	30.39	7.3	19.86	82	1.11	
VEGAN dinner	444	15.4	20.19	55.62	16.3	13.74	176	3.5	
VEGAN snack #3	378	7.41	18.69	50.46	6.8	23.42	59	3.02	
Dietary
pattern/meal	Sodium,
Na (mg)	Vitamin C,
total ascorbic
acid (mg)	Cholesterol
(mg)	Carotene,
beta (mcg)	Sucrose
(g)	Glucose
(dextrose)
(g)	Fructose
(g)	Lactose
(g)	
AMERICAN breakfast	300	0	17	0	N/A	N/A	N/A	N/A	
AMERICAN lunch	1,399	12.1	79	0	1.27	2.28	3.7	0.7	
AMERICAN dinner	2,102	7.2	56	N/A	N/A	N/A	N/A	N/A	
AMERICAN snack	160	0	0	N/A	N/A	N/A	N/A	N/A	
USDA breakfast	278	16.1	12	61	0.12	1.14	1.45	12.69	
USDA snack #1	100	8	10	47	3.08	3.62	8.79	0	
USDA lunch	1,516	110	223	56	0.15	7.2	8.13	0	
USDA snack #2	863	3.9	23	9,600	4.08	3.3	1.5	6.55	
USDA dinner	446	25.6	90	1,953	3.78	1.58	1.56	0	
VEGAN breakfast	312	6	85	18	3.14	8.49	9.03	0	
VEGAN snack #1	266	80	0	69	1.79	10.47	10.51	0	
VEGAN lunch	1,618	16.2	0	6,850	2.65	1.4	1.11	0	
VEGAN snack #2	9	8.4	0	50	4.63	4.45	10.76	0	
VEGAN dinner	499	102.3	0	1,018	0.11	6.96	3.23	0.19	
VEGAN snack #3	169	1.6	0	17	1.12	0.03	0.03	0	
Dietary
pattern/meal	Maltose
(g)	Galactose
(g)	Starch
(g)	Fatty acids, total
monounsaturated
(g)	Fatty acids, total
polyunsaturated
(g)	Vitamin A,
IU (IU)	Fatty acids, total
saturated (g)	
AMERICAN breakfast	N/A	N/A	N/A	N/A	N/A	0	2.64	
AMERICAN lunch	1.07	0	90.32	19.63	7.86	412	11.52	
AMERICAN dinner	N/A	N/A	N/A	N/A	N/A	600	11.4	
AMERICAN snack	N/A	N/A	N/A	N/A	N/A	0	2	
USDA breakfast	0	0	0	0.72	0.33	499	1.97	
USDA snack #1	0	0	0.07	1.16	0.55	836	2.06	
USDA lunch	0	0	0	7.71	11.43	5,179	7.29	
USDA snack #2	0	0	0	1.01	0.27	20,852	5.99	
USDA dinner	0.05	0.02	2.79	9.88	1.61	3,271	1.97	
VEGAN breakfast	0.21	0.43	4.43	3.9	2.32	544	2.25	
VEGAN snack #1	0.01	0	3.7	1.06	2.6	5,129	0.62	
VEGAN lunch	0	0.29	0.87	5.9	5.82	13,184	1.79	
VEGAN snack #2	0.01	0.01	0.27	7.99	3.48	102	1.48	
VEGAN dinner	0.19	0.01	0	10.74	5.56	1,779	2.64	
VEGAN snack #3	0	0	8.81	12.48	3.51	52	1.69	

The following meals contained fermented foods that contained live active cultures according to the package and were prepared without heat treatment: USDA meal plan snack #1 (yogurt), lunch (parmesan cheese), and snack #2 (cottage cheese). The following meals contained fermented foods that were cooked as part of meal preparation: VEGAN meal plan lunch (tofu), and AMERICAN meal plan lunch and dinner (cheese). Meal ingredients were purchased at local grocery stores in Saint Helena, CA, and prepared meals were purchased in restaurants in Napa, CA.

Plate counts

The aerobic, anaerobic, yeast and mold plate counts are shown in Table 1. The meals ranged in total numbers of microorganisms from CFU to CFU with the aerobic and anaerobic plate counts being among the highest and the yeast and mold plate counts being among the lowest across all meals. The USDA dietary pattern had the highest total microorganisms for the day at CFU mostly due to the higher amounts of anaerobic bacteria in the morning snack (CFU) and higher amounts of aerobic and anaerobic bacteria in the afternoon snack (5.5 × 108 and 6 × 108 CFU respectively). Not surprisingly, both of these meals contained fermented products, in the first case yogurt, and in the second case cottage cheese. The AMERICAN and VEGAN dietary patterns had 3 orders of magnitude fewer total microorganisms than the USDA dietary patterns, with total microorganisms of CFU and CFU respectively. Neither the AMERICAN nor the VEGAN dietary pattern meals contained fermented foods that were not heat treated as part of meal preparation. The AMERICAN lunch and dinner contained cheese that was either cooked on a grill or baked in the oven and the VEGAN lunch contained tofu, which was cooked in the vegetable broth. The USDA lunch also had the highest amounts of yeast and mold (and CFU respectively) of all the meals, and this meal also had relatively high amounts of aerobic bacteria (CFU). In the VEGAN dietary pattern, the morning snack had the highest amounts of aerobic and anaerobic bacteria (and CFU respectively).

Sequence processing and summary statistics

The number of high-quality sequences per sample (i.e., meal) ranged from 168,669 to 318,956 (see Table 6). Sequences were clustered and clusters were assigned to a taxonomic group (when possible) using the pick_open_reference_otus.py script with a 97% similarity cutoff and the gg_13_8_otus reference taxonomy provided by the Greengenes Database Consortium (http://greengenes.secondgenome.com). After OTU assignment, mitochondrial and chloroplast sequences were filtered out, sequences that were observed only once across all samples were removed, and sequences that were Unassigned at the Domain taxonomic level were removed (these Unassigned sequences were verified via a manual BLAST search to be chloroplast sequences). After this filtration, the range of sequences per sample decreased to 771–244,597. All subsequent beta diversity analyses (comparisons across samples) were performed on samples that were rarefied to 771 sequences per sample.

Table 6 Sequence summary statistics by meal.

The number of sequences per meal before and after filtration to remove eukaryotic, chimeric, and singleton reads, and the number of OTUs per meal after filtration.

Meal	# Sequences
pre-filtration	# Sequences
post-filtration	# OTUs (open reference,
97% similarity)	
AMERICAN breakfast	267,254	226,903	1,838	
AMERICAN dinner	298,442	11,666	660	
AMERICAN lunch	299,035	96,898	622	
AMERICAN snack	311,311	279,136	969	
USDA breakfast	318,956	5,002	502	
USDA dinner	277,213	6,149	476	
USDA lunch	270,166	16,456	607	
USDA snack1	299,998	226,403	334	
USDA snack2	238,057	104,114	333	
VEGAN breakfast	274,360	7,310	399	
VEGAN dinner	303,246	3,576	417	
VEGAN lunch	291,459	13,874	480	
VEGAN snack1	244,886	62,446	644	
VEGAN snack2	288,319	974	1,053	
VEGAN snack3	168,669	54,483	229	

Taxonomic composition and diversity of the different dietary patterns

In terms of taxonomic alpha diversity, there was no significant difference between dietary patterns (Fig. 2) (non-parametric Kruskal–Wallis test with compare_alpha_diversity.py, p > 0.6). This is the case for multiple diversity metrics, including a count of the absolute number of OTUs observed, as well as the Chao1 and Shannon–Weiner (parametric and nonparametric, respectively) diversity indices, which account for the relative abundance (evenness) of the OTUs observed. We also tested for the significant variation in frequency of individual OTUs between diet types using the Kruskal–Wallis test, as implemented in the group_significance.py script. This test is appropriate for comparing independent groups, with unequal sample sizes, that may not be normally distributed. None of the OTUs were significantly different between the three diet types. The most abundant 50 OTUs (clustered at 97% similarity) belong to 25 different bacterial families, including many that are commonly found in association with plants and animals (see Fig. 3).

Figure 2 Alpha diversity measures for the three diet types.

While some individual meals had higher alpha diversity (defined either by the number of OTUs observed or by the Chao1 and Shannon diversity measures) than others, there was no significant difference in diversity between the different dietary patterns (AMERICAN, USDA, and VEGAN).

Figure 3 The cumulative relative abundance of Families representing the 50 most abundant OTUs.

The 50 most abundant OTUs in this study (clustered at 97% similarity) belong to 25 different bacterial families, including many that are commonly found in association with plants and animals. None of them vary significantly with respect to diet type.

Factors driving the differences in microbial community composition and diversity of individual meals

There was no effect of dietary pattern on the overall community composition within individual meals (PERMANOVA with compare_categories.py, p = 0.591). There was no obvious clustering based on any potentially distinguishing feature tested, including whether the meals contained fermented foods, dairy, whether they ware raw or cooked, or the calculated nutritional content (see Table 5 for complete meal metadata). However, different meals clustered together independent of dietary pattern. For example, meals that were relatively abundant in Prevotellaceae included the USDA dinner, VEGAN dinner, AMERICAN dinner, USDA breakfast, VEGAN snack 2, and VEGAN snack 3. Prevotellaceae includes organisms that tend to be very abundant in the guts of many animals, and have been associated with Inflammatory Bowel Disease in humans (Henderson et al., 2013; Wu, Bushmanc & Lewis, 2013). The AMERICAN snack, AMERICAN lunch, USDA snack 2 and USDA snack 1 had a high relative abundance of Streptococcaceae (Fig. 4). It is difficult to know what specific features of these meals made them similar in this regard. Possible contributing factors may be provenance of ingredients and/or individual meal components such as presence of a certain fruit or vegetable.

Figure 4 Heatmap of taxa abundance in each meal.

Heatmap showing relative abundance of bacterial families of individual meals. Similarities between meals are not necessarily part of the same dietary pattern. Hierarchical clustering is based on Ward clustering of the Pearson correlation coefficients, with sample by sample normalization performed using the median.

A large amount of variation (52%) was explained by PCo1 (i.e., the eigenvector that explains the most variation)(Fig. 1), and our attempts to determine the factors driving this variation led us to look at specific taxonomic groups that may be important. We did this in two ways. First, we produced a biplot with the make_emperor.py script, showing the prevalence of bacterial families in the PCoA space defined by the weighted unifrac distance between the 15 meals (Fig. 5). A cluster of 4 meals, including USDA snack 1, AMERICAN snack, AMERICAN lunch, and USDA lunch, was comprised of samples that were dominated by Lactic Acid Bacteria. These are members of the order Lactobacillales, which are commonly found in association with both food products, especially in fermented milk products (Terzic-Vidojevic et al., 2014) and human mucosal surfaces (Rizzello et al., 2011). The Vegan snack #1 was unique in that it was dominated (70.4%) by Xanthamonadaceae, a family containing many plant pathogens. A second cluster of 7 meals including VEGAN dinner, VEGAN breakfast, AMERICAN breakfast, AMERICAN dinner, USDA dinner, VEGAN snack 3, and USDA breakfast, was comprised of samples containing a large percentage (average = 27%) of Thermus, a clade with many heat and dessication-resistant organisms. Second, we calculated correlations between the relative abundance of a single taxon and the PCo1 value for each meal using a simple regression (see Table 7). The bacterial family most tightly correlated with PCo1 was Streptococcaceae (r = 0.852).

Figure 5 Biplot of taxa in sample PCoA space.

Bacterial families (light blue spheres) are displayed in a PCoA biplot based on weighted Unifrac distances between the 15 meals. The size of the spheres representing taxa is correlated with the relative abundance of the labeled organism. In the interest of readability, only the bacterial families discussed in the text are labeled. Axes are scaled to amount of variation explained.

Table 7 Pairwise Pearson correlations between individual taxonomic groups and meal nutrient composition.

Pairwise Pearson correlation coefficients (R) reveal significant correlations between some taxonomic groups and meal nutrient contents. Correlations were performed at 5 taxonomic levels (Phylum-Genus.) Only significant correlations are reported here.

OTU taxonomy string	Energy	Protein	Total lipid	Carbohydrate	Fiber	Sugars	Calcium	
k__Bacteria
p__Proteobacteria	−0.0555	0.1469	−0.1615	0.0001	0.21	0.1548	0.306	
k__Bacteria
p__Proteobacteria
c__Betaproteobacteria	0.1243	0.2057	0.2017	−0.0193	0.0075	−0.1199	−0.0134	
k__Bacteria
p__Proteobacteria
c__Gammaproteobacteria	−0.0518	0.1531	−0.1693	0.0122	0.2062	0.1612	0.3298	
k__Bacteria
p__Proteobacteria
c__Betaproteobacteria
o__Burkholderiales	0.1244	0.2036	0.2057	−0.0206	0.0177	−0.1158	−0.0207	
k__Bacteria
p__Proteobacteria
c__Gammaproteobacteria
o__Aeromonadales	0.1881	0.3457	0.2304	0.0074	−0.0007	−0.246	0.0846	
k__Bacteria
p__Proteobacteria
c__Gammaproteobacteria
o__Pseudomonadales	0.2104	0.3438	0.2531	0.0501	0.1325	−0.1955	0.1255	
k__Bacteria
p__Bacteroidetes
c__Bacteroidia
o__Bacteroidales
f__[Odoribacteraceae]	0.0254	0.2495	0.0801	−0.1591	0.4575	−0.4456	−0.5145*	
k__Bacteria
p__Firmicutes
c__Bacilli
o__Bacillales
f__Staphylococcaceae	−0.1049	0.0316	−0.0643	−0.1813	0.2204	−0.5232*	−0.1309	
k__Bacteria
p__Proteobacteria
c__Betaproteobacteria
o__Burkholderiales
f__Oxalobacteraceae	0.1536	0.2519	0.2072	0.0201	−0.023	−0.0741	0.0677	
k__Bacteria
p__Proteobacteria
c__Gammaproteobacteria
o__Pseudomonadales
f__Pseudomonadaceae	0.2383	0.3882	0.2629	0.0766	0.1229	−0.1943	0.1457	
k__Bacteria
p__Firmicutes
c__Bacilli
o__Bacillales
f__Staphylococcaceae
g__Staphylococcus	−0.1031	0.0343	−0.0633	−0.1798	0.222	−0.5254*	−0.1292	
k__Bacteria
p__Firmicutes
c__Bacilli
o__Lactobacillales
f__Streptococcaceae
g__Lactococcus	0.2088	0.2213	0.2812	0.0331	0.1385	0.142	−0.0006	
k__Bacteria
p__Firmicutes
c__Clostridia
o__Clostridiales
f__Lachnospiraceae
g__Blautia	−0.0198	0.1057	0.028	−0.1283	0.3711	−0.5586*	−0.2554	
k__Bacteria
p__Proteobacteria
c__Gammaproteobacteria
o__Pseudomonadales
f__Pseudomonadaceae
g__Pseudomonas	0.2379	0.3875	0.2629	0.0764	0.1225	−0.1935	0.1452	
OTU taxonomy string	Iron	Sodium	Vitamin C	Vitamin A	Fatty acids	Cholesterol	beta Carotene	
k__Bacteria
p__Proteobacteria	0.5187*	−0.061	0.576*	−0.0015	−0.2234	0.026	−0.2217	
k__Bacteria
p__Proteobacteria
c__Betaproteobacteria	0.1572	0.2141	0.4799	−0.1432	0.1704	0.731**	−0.2859	
k__Bacteria
p__Proteobacteria
c__Gammaproteobacteria	0.5275*	−0.0574	0.5611*	0.0223	−0.2227	0	−0.1961	
k__Bacteria
p__Proteobacteria
c__Betaproteobacteria
o__Burkholderiales	0.176	0.2094	0.5029	−0.1219	0.1636	0.7205**	−0.2679	
k__Bacteria
p__Proteobacteria
c__Gammaproteobacteria
o__Aeromonadales	0.2563	0.4289	0.5015	−0.0596	0.3302	0.7054**	−0.2024	
k__Bacteria
p__Proteobacteria
c__Gammaproteobacteria
o__Pseudomonadales	0.3722	0.3475	0.6911**	0.0597	0.2116	0.7658**	−0.1142	
k__Bacteria
p__Bacteroidetes
c__Bacteroidia
o__Bacteroidales
f__[Odoribacteraceae]	−0.0743	−0.1711	0.2335	−0.2165	−0.2105	−0.0428	−0.0989	
k__Bacteria
p__Firmicutes
c__Bacilli
o__Bacillales
f__Staphylococcaceae	0.1455	0.1666	−0.0834	0.005	−0.2458	−0.1784	0.142	
k__Bacteria
p__Proteobacteria
c__Betaproteobacteria
o__Burkholderiales
f__Oxalobacteraceae	0.2238	0.2378	0.5455*	−0.0402	0.1842	0.7828**	−0.2121	
k__Bacteria
p__Proteobacteria
c__Gammaproteobacteria
o__Pseudomonadales
f__Pseudomonadaceae	0.4155	0.3834	0.7115**	0.0968	0.2367	0.764**	−0.0822	
k__Bacteria
p__Firmicutes
c__Bacilli
o__Bacillales
f__Staphylococcaceae
g__Staphylococcus	0.1482	0.1686	−0.0819	0.0068	−0.2455	−0.1771	0.1437	
k__Bacteria
p__Firmicutes
c__Bacilli
o__Lactobacillales
f__Streptococcaceae
g__Lactococcus	−0.0765	0.136	−0.2465	0.603*	0.3762	−0.0024	0.6236*	
k__Bacteria
p__Firmicutes
c__Clostridia
o__Clostridiales
f__Lachnospiraceae
g__Blautia	0.2255	0.1582	0.0478	0.0257	−0.2386	−0.2277	0.1791	
k__Bacteria
p__Proteobacteria
c__Gammaproteobacteria
o__Pseudomonadales
f__Pseudomonadaceae
g__Pseudomonas	0.4149	0.3819	0.7113**	0.0964	0.236	0.7645**	−0.0828	
Notes.

* p < 0.05.

** p < 0.01.

We also asked whether the relative abundance of any particular taxonomic group was correlated with the nutritional content of the meals via pairwise Pearson’s correlations. We limited this analysis to organisms that were present in all 15 meals. Due to the exploratory nature of this study, there were no specific hypotheses tested with these correlations, and therefore no corrections for multiple hypothesis testing. Some taxa frequently abundant in human microbiome studies were found to be significantly correlated with particular nutrients (Table 7). For example, members of the genus Blautia are frequently observed in human fecal samples, and in our study, the relative abundance of this genus was found to be positively correlated with the sugar content of the meals p < 0.05 (Fig. 6). We emphasize here that due to the large numbers of OTUs present in this study, corrected p values were always non-significant. However, the goal of this small-scale study is to inform the development of future hypotheses, not test current ones. Nevertheless, this result suggests that there could be interesting relationships between the nutritional content of the foods that we eat, the microbes that associate with those foods, and our gut microbiome, not just because we are “feeding” our gut microbes, but because we are eating them as well (but, see “Caveats” section below.)

Figure 6 Correlation of Blautia abundance with sugar content in meals.

Scatterplot with simple regression line of the relative abundance of Blautia versus grams of sugar in each sample (i.e., meal). Pearson’s r = 0.56.

Metagenome prediction with PICRUSt

Because of the vast, historical effort to make the 16S rRNA gene sequence available for hundreds of thousands of organisms, we are typically able to characterize well the taxonomic diversity of most microbial communities. One might assume that each organism present in a community has some functional role to play, and the most straightforward way to predict what that role each organism might play is to use metagenomic sequencing to interrogate the genomes of all members of the community. Unfortunately, in many cases and with current sequencing technology, the amount of microbial DNA relative to host or other environmental DNA is small enough to make metagenomic sequencing infeasible. This is the case here, where the plant and animal DNA present in the food we eat is typically much more abundant than the microbial DNA. Some exceptions may exist with respect to fermented foods, but we are equally interested in the microbiota associated with a wide variety of food types.

In a case like this for which metagenomic sequencing is infeasible, another approach suggests itself. There is evidence that a correlation exists between the evolutionary relatedness of two organisms and the similarity of their genomic content (Martiny, Treseder & Pusch, 2012). This allows us to leverage the information obtained by sequencing the genome of one organism to predict the functional potential of another, even if the other genome is represented only by a 16S rRNA sequence. The power of this approach is increased when very many, very closely-related genome sequences are available. This predictive approach has recently been implemented in the software package PICRUSt. PICRUSt uses the phylogenetic placement of a 16S rRNA sequence within a phylogeny of sequenced genomes to infer the content of the genome of the organism represented by that 16S rRNA sequence.

With PICRUSt one can calculate a metric (NSTI) that measures how closely related the average 16S rDNA sequence in an environmental sample is to an available sequenced genome. When this number is low, PICRUSt is likely to perform well in predicting the genomes of the organisms in an environmental sample (i.e., a metagenome). The average NSTI for our 15 meals was 0.038, which is on par with the NSTI for the Human Microbiome Project samples (mean NSTI = 0.03 ± 0.02 s.d.), for which a massive effort has been made to obtain reference genome sequences (Proctor, 2011). This low NSTI metric suggests that PICRUSt may perform well when predicting the metabolic potential of the microbial communities found in the meals prepared for this study. Here, we have shown the most significant KEGG functional category, for “Other N-glycan degradation” (KO 00511, p = 8.21e−3), which was highest in the VEGAN dietary pattern (Fig. 7). Again, this is not a significant result when a p-value correction is applied, but is nevertheless highlighted as a potential source of information when using a pilot study like this to inform future research questions. As a sanity check for the PICRUSt predictions, we compared the relative abundance of genes present in the KEGG functional category “Sporulation” between meals that were cooked were compared to those that were raw (Fig. 8). As expected, because organisms that can form spores are more likely to survive the cooking process, Sporulation-associated genes are more abundant in cooked versus raw foods. All KEGG (Level 3) pathways that vary significantly between dietary patterns are presented in Table 8. These findings suggest that there are functional differences in bacterial populations associated with different foods and meals, and that these may be related not only to bacterial substrate preferences, but also techniques used in meal preparation.

Figure 7 PICRUSt metagenome prediction suggests higher abundance of genes in the “Other glycan degradation” KEGG pathway in the VEGAN diet.

Metagenome prediction with PICRUSt reveals functional categories that differ significantly between the AMERICAN, USDA, and VEGAN diet types. The abundance of genes annotated in the “Other glycan degradation” (KO00155) pathway are significantly higher in the VEGAN diet (p = 8.21e−3). Due to the exploratory nature of this data set corrections for multiple testing were not applied.

Figure 8 PICRUSt metagenome prediction suggests higher abundance of genes in the “Sporulation” KEGG pathway in cooked meals.

Metagenome prediction with PICRUSt reveals functional categories that differ significantly between the cooked and raw meal types. The abundance of genes annotated in the “Sporulation” pathway are significantly higher in the cooked meals (p = 0.039). Due to the exploratory nature of this data set corrections for multiple testing were not applied.

Table 8 KEGG Pathways with significant differences between meal categories as predicted by PICRUSt.

PICRUSt was used to predict the functional potential of the microbial community found in each meal. This table contains all of the predicted KEGG pathways (at the 3rd hierarchical level) that vary significantly (p-value <0.05) across nutrient composition or meal category type.

Nutrient/Descriptor	KEGG functional category (level 3)	Type of test used	p-value	
Calcium_bin	Peptidases	ANOVA	0.039	
Carotene_beta_bin	Prenytransferases	Welch’s T-test	0.041	
Carotene_beta_bin	Vibrio cholera pathogenic cycle	0.041	0.042	
Dairy	Other glycan degradation	Welch’s T-test	0.041	
Protein	Phenypropanoid biosynthesis	ANOVA	0.021	
Vitamin_C_bin	Calcium signaling pathway	ANOVA	0.023	
Vitamin_C_bin	Transporters	ANOVA	0.049	
Vitamin_A	Cytoskeleton proteins	ANOVA	0.017451421	
Vitamin_A	Peptidases	ANOVA	0.019130989	
Vitamin_A	Flavonoid biosynthesis	ANOVA	0.023972731	
Vitamin_A	Germination	ANOVA	0.029007085	
Vitamin_A	Chaperones and folding catalysts	ANOVA	0.031168211	
Total/lipid/bin	Secondary bile acid biosynthesis	ANOVA	0.029181829	
Total/lipid/bin	Biosynthesis of siderophore group nonribosomal peptides	ANOVA	0.043392615	
Sodium	Peptidases	ANOVA	0.014721635	
Sodium	Benzoate degradation	ANOVA	0.019865137	
Sodium	Limonene and pinene degradation	ANOVA	0.031164989	
Sodium	Butanoate metabolism	ANOVA	0.032537666	
Sodium	Nucleotide excision repair	ANOVA	0.033766923	
Sodium	Phenylalanine, tyrosine and
tryptophan biosynthesis	ANOVA	0.036768304	
Sodium	Peroxisome	ANOVA	0.037176174	
Sodium	Ethylbenzene degradation	ANOVA	0.039923012	
Sodium	Naphthalene degradation	ANOVA	0.040749468	
Sodium	Restriction enzyme	ANOVA	0.043486893	
Sodium	Tyrosine metabolism	ANOVA	0.047200716	
Iron_bin	Carbon fixation in photosynthetic organisms	ANOVA	0.021668245	
Iron_bin	Protein kinases	ANOVA	0.022263474	
Iron_bin	Translation proteins	ANOVA	0.024669694	
Iron_bin	Pyruvate metabolism	ANOVA	0.032434551	
Iron_bin	Thiamine metabolism	ANOVA	0.038685974	
Iron_bin	D-Glutamine and D-glutamate metabolism	ANOVA	0.041514757	
Iron_bin	One carbon pool by folate	ANOVA	0.042928705	
Fiber_bin	Other glycan degradation	ANOVA	0.014660951	
Fiber_bin	N-Glycan biosynthesis	ANOVA	0.042139092	
Fiber_bin	Proteasome	ANOVA	0.047590993	
Fiber_bin	Prostate cancer	ANOVA	0.048468144	
Fiber_bin	Antigen processing and
presentation	ANOVA	0.048618504	
Fiber_bin	Progesterone-mediated oocyte maturation	ANOVA	0.048618504	
Fiber_bin	Other transporters	ANOVA	0.049130295	
Fermented	Transcription factors	Welch’s T-test	0.009468352	
Fermented	Phosphonate and phosphinate metabolism	Welch’s T-test	0.011059743	
Fermented	Cytoskeleton proteins	Welch’s T-test	0.011669915	
Fermented	Amoebiasis	Welch’s T-test	0.012857954	
Fermented	Oxidative phosphorylation	Welch’s T-test	0.013841012	
Fermented	Transporters	Welch’s T-test	0.015647729	
Fermented	Protein processing in endoplasmic reticulum	Welch’s T-test	0.017210951	
Fermented	Riboflavin metabolism	Welch’s T-test	0.017938599	
Fermented	Steroid hormone biosynthesis	Welch’s T-test	0.020976432	
Fermented	PPAR signaling pathway	Welch’s T-test	0.021710104	
Fermented	Peroxisome	Welch’s T-test	0.02186366	
Fermented	Citrate cycle (TCA cycle)	Welch’s T-test	0.021867149	
Fermented	Toluene degradation	Welch’s T-test	0.022788506	
Fermented	Carbon fixation pathways in prokaryotes	Welch’s T-test	0.024411154	
Fermented	alpha-Linolenic acid metabolism	Welch’s T-test	0.026939926	
Fermented	Methane metabolism	Welch’s T-test	0.030552388	
Fermented	Synthesis and degradation of ketone bodies	Welch’s T-test	0.031611873	
Fermented	RNA degradation	Welch’s T-test	0.032553878	
Fermented	Dioxin degradation	Welch’s T-test	0.036756638	
Fermented	Adipocytokine signaling pathway	Welch’s T-test	0.04018659	
Fermented	Benzoate degradation	Welch’s T-test	0.040925365	
Fermented	Chlorocyclohexane and
chlorobenzene degradation	Welch’s T-test	0.04187257	
Fermented	Nicotinate and nicotinamide metabolism	Welch’s T-test	0.049217654	
Fatty_acids_bin	Other glycan degradation	ANOVA	0.002774493	
Fatty_acids_bin	N-Glycan biosynthesis	ANOVA	0.009783852	
Fatty_acids_bin	Proteasome	ANOVA	0.012010853	
Fatty_acids_bin	Prostate cancer	ANOVA	0.014268966	
Fatty_acids_bin	Antigen processing and presentation	ANOVA	0.014432811	
Fatty_acids_bin	Progesterone-mediated oocyte maturation	ANOVA	0.014432811	
Fatty_acids_bin	NOD-like receptor signaling pathway	ANOVA	0.019075184	
Fatty_acids_bin	Chloroalkane and chloroalkene degradation	ANOVA	0.02167892	
Fatty_acids_bin	Taurine and hypotaurine metabolism	ANOVA	0.022063389	
Fatty_acids_bin	Sphingolipid metabolism	ANOVA	0.028794158	
Fatty_acids_bin	Other transporters	ANOVA	0.033398699	
Fatty_acids_bin	Primary bile acid biosynthesis	ANOVA	0.033705405	
Fatty_acids_bin	Stilbenoid, diarylheptanoid and gingerol biosynthesis	ANOVA	0.034295974	
Fatty_acids_bin	Glycolysis/Gluconeogenesis	ANOVA	0.041601128	
Fatty_acids_bin	Amyotrophic lateral sclerosis (ALS)	ANOVA	0.045157824	
Fatty_acids_bin	Purine metabolism	ANOVA	0.048100832	
Energy_bin	Nucleotide excision repair	ANOVA	0.002532996	
Energy_bin	Chromosome	ANOVA	0.003249937	
Energy_bin	Ubiquinone and other
terpenoid-quinone
biosynthesis	ANOVA	0.00500867	
Energy_bin	Mismatch repair	ANOVA	0.006531889	
Energy_bin	Photosynthesis proteins	ANOVA	0.00816459	
Energy_bin	Photosynthesis	ANOVA	0.009638124	
Energy_bin	Restriction enzyme	ANOVA	0.009805668	
Energy_bin	Carbohydrate metabolism	ANOVA	0.009966615	
Energy_bin	Limonene and pinene degradation	ANOVA	0.010243464	
Energy_bin	DNA replication proteins	ANOVA	0.010451223	
Energy_bin	Lipoic acid metabolism	ANOVA	0.010843214	
Energy_bin	Phenylalanine, tyrosine and
tryptophan biosynthesis	ANOVA	0.011059994	
Energy_bin	Peptidases	ANOVA	0.01407109	
Energy_bin	DNA repair and recombination proteins	ANOVA	0.015827001	
Energy_bin	Type II diabetes mellitus	ANOVA	0.016276453	
Energy_bin	alpha-Linolenic acid metabolism	ANOVA	0.018912274	
Energy_bin	Butanoate metabolism	ANOVA	0.020566276	
Energy_bin	Homologous recombination	ANOVA	0.025440256	
Energy_bin	Flavone and flavonol biosynthesis	ANOVA	0.025662772	
Energy_bin	Protein export	ANOVA	0.025889019	
Energy_bin	DNA replication	ANOVA	0.026934029	
Energy_bin	Primary immunodeficiency	ANOVA	0.027524778	
Energy_bin	Glycosphingolipid biosynthesis—
lacto and neolacto series	ANOVA	0.027782212	
Energy_bin	Indole alkaloid biosynthesis	ANOVA	0.028725346	
Energy_bin	Amoebiasis	ANOVA	0.028866801	
Energy_bin	Benzoate degradation	ANOVA	0.029913338	
Energy_bin	D-Alanine metabolism	ANOVA	0.029961727	
Energy_bin	C5-Branched dibasic acid metabolism	ANOVA	0.030293739	
Energy_bin	Peptidoglycan biosynthesis	ANOVA	0.031023499	
Energy_bin	Glycerolipid metabolism	ANOVA	0.032020201	
Energy_bin	Bisphenol degradation	ANOVA	0.032129248	
Energy_bin	Betalain biosynthesis	ANOVA	0.032513208	
Energy_bin	Biosynthesis of siderophore group nonribosomal peptides	ANOVA	0.033939326	
Energy_bin	Melanogenesis	ANOVA	0.035235651	
Energy_bin	Amino acid metabolism	ANOVA	0.037736626	
Energy_bin	Ribosome Biogenesis	ANOVA	0.039545196	
Energy_bin	Peroxisome	ANOVA	0.040442991	
Energy_bin	Steroid hormone biosynthesis	ANOVA	0.041078867	
Energy_bin	Amino sugar and nucleotide sugar metabolism	ANOVA	0.042902641	
Energy_bin	Phosphotransferase system (PTS)	ANOVA	0.043431781	
Energy_bin	Arginine and proline metabolism	ANOVA	0.043807314	
Energy_bin	Glycine, serine and threonine metabolism	ANOVA	0.044386803	
Energy_bin	Riboflavin metabolism	ANOVA	0.044908782	
Energy_bin	Metabolism of cofactors
and vitamins	ANOVA	0.04499208	
Energy_bin	Systemic lupus erythematosus	ANOVA	0.045791434	
Energy_bin	Biosynthesis of type II polyketide products	ANOVA	0.047100884	
DietType	Other glycan degradation	ANOVA	0.007486572	
DietType	N-Glycan biosynthesis	ANOVA	0.03160732	
DietType	Proteasome	ANOVA	0.034750295	
DietType	Prostate cancer	ANOVA	0.03926696	
DietType	Antigen processing and presentation	ANOVA	0.039372465	
DietType	Progesterone-mediated oocyte maturation	ANOVA	0.039372465	
Cooked	mRNA surveillance pathway	Welch’s T-test	0.006445676	
Cooked	Cell cycle	Welch’s T-test	0.009317081	
Cooked	Hepatitis C	Welch’s T-test	0.009317081	
Cooked	Measles	Welch’s T-test	0.009317081	
Cooked	mTOR signaling pathway	Welch’s T-test	0.009317081	
Cooked	Phagosome	Welch’s T-test	0.009317081	
Cooked	Transcription machinery	Welch’s T-test	0.021347218	
Cooked	Various types of N-glycan
biosynthesis	Welch’s T-test	0.023199289	
Cooked	Sporulation	Welch’s T-test	0.024693602	
Cooked	Vibrio cholerae infection	Welch’s T-test	0.025037566	
Cooked	Cytoskeleton proteins	Welch’s T-test	0.04573813	
Cooked	Cytochrome P450	Welch’s T-test	0.049095188	
Cholesterol_bin	Transcription machinery	ANOVA	0.007389033	
Cholesterol_bin	Plant-pathogen interaction	ANOVA	0.010583548	
Cholesterol_bin	Folate biosynthesis	ANOVA	0.012498951	
Cholesterol_bin	Tetracycline biosynthesis	ANOVA	0.034405715	
Cholesterol_bin	Other ion-coupled transporters	ANOVA	0.036332221	
Cholesterol_bin	Proteasome	ANOVA	0.039954688	
Cholesterol_bin	Valine, leucine and isoleucine biosynthesis	ANOVA	0.040216227	
Cholesterol_bin	General function prediction only	ANOVA	0.042978367	
Cholesterol_bin	Prostate cancer	ANOVA	0.044525651	
Cholesterol_bin	Antigen processing and presentation	ANOVA	0.044753096	
Cholesterol_bin	Progesterone-mediated oocyte maturation	ANOVA	0.044753096	
Cholesterol_bin	NOD-like receptor signaling pathway	ANOVA	0.047390633	

Summary

In this study we estimated the total numbers and kinds of microorganisms consumed in a day by an average American adult. We analyzed meals representing three typical dietary patterns, including the Average American, USDA recommended, and Vegan diet, and found that Americans likely consume in the range of 106–109 CFUs microbes per day. The USDA meal plan included two meals with non-heat treated fermented foods, which were likely responsible for the 3-fold higher total microbes in this meal plan compared to the AMERICAN and VEGAN diets. Food preparation techniques such as heating or acid treatment can kill bacteria, however, these processes affect different bacteria to different degrees. For example, spore-forming bacteria can survive heat treatment (Stringer, Webb & Peck, 2011) (also see Fig. 8). Once inside the gastrointestinal tract, the low pH of the stomach, as well as bile salts also kill some bacteria, but not those that are acid and/or bile salt resistant. It is unknown what proportion of the microbes we eat make it through the hostile environment of the gastrointestinal tract. However, a recent study showed that food microbes consumed as part of fermented foods such as cheese did appear in the stool and were culturable (David et al., 2013).

We also used PICRUSt to predict the functional potential of the microbiota associated with each meal in this study. Of course, this is not a perfect substitute for metagenomic sequencing or experimental studies, but it does allow one to develop some initial hypotheses related to the function of a microbial community. For example, between diet types, the most significant difference in KEGG functional categories was for “other N-glycan degradation”. This function was over-represented in the Vegan diet, which is perhaps not surprising given that cellulose is a glycan, and the Vegan diet is significantly higher in cellulose than the others. This suggests that when one consumes a diet that is high in cellulose, one also consumes a population of microbes that is well equipped to digest cellulose.

Caveats

It is important to point out some caveats with regard to this study. First, the scale of this study was limited. Our objective with this preliminary study was to explore the possibility that the microbes in our food may be meaningful contributors to the ecosystem of our gut microbiota. The current paradigm is that the harsh conditions of the human gastrointestinal tract (i.e., high acidity in the stomach, presence of bile acids and digestive enzymes in the small intestine) preclude most microbes present in and on food from playing significant roles in the gastrointestinal microbial ecosystem because they do not survive intact. Yet recent reports show that there are microbial blooms within 24 h of large shifts in the diet (Wu et al., 2011; Walker et al., 2011), attributed to changes in the available fermentable substrate, which in turn promoted the growth of specific bacteria already present in the gut. The possibility remains that microbes present on the food itself contributed at least in part to these observed transient blooms or shifts. Very little is known about how different food matrices may promote the survival of food bacteria despite their lack of acid and bile acid resistance in vitro. Transient shifts in gut microbiome composition caused by dietary factors may be unimportant when the gut microbiome rebounds to its initial state. However, it is possible that under certain conditions, such perturbations could lead to functional and long-lasting changes. The goal of this study was not to explore the explicit effects of ingesting food microbes on changes in the gut microbiota but simply to provide preliminary data on the composition and numbers of bacteria in typical American dietary patterns.

Second, the study did not aim to be exhaustive in its exploration of diets. We did not aim to produce statistically significant differences in microbial composition and quantity in replicates across multiple days of a particular dietary pattern. Instead, the aim was to generate hypotheses about dietary microbes and their variation across meals and diets that can now be followed up with more rigorous studies. Because the purpose of this study was to generate hypotheses rather than test specific hypotheses, multiple testing corrections were not applied in our statistical analyses.

Third, the microbial counts reported here are rough estimates of the total amount of microbes consumed in a day by an average American eating meals described here. It is important to note that inherent to the plate count techniques used, not every microbe will grow under these culture conditions, and the plate counts are only estimates.

Future directions

It is possible that part of the high variation in gut microbiota composition observed among individuals is due to the specific and complex differences in diet beyond the nutrient composition that can be estimated from dietary records and recalls. This study begs the question: do the microbes we eat as part of our normal daily diets contribute to the composition and function of our gut microbiota? There are many questions that remain to be answered. Under what circumstances do microbes consumed as part of meals remain in the gastrointestinal tract transiently versus persistently following a meal? Do the microbes we eat affect the function of the resident gut microbiota, even if they do not affect its composition, as has been suggested by some yogurt feeding studies (McNulty et al., 2011)? How do different cooking and preparation methods affect the microbial composition of meals and the survival characteristics of individual microbes through the gastrointestinal tract? How do specific factors such as length of transport or provenance of individual ingredients (e.g., imported vs. domestic), packaging materials, and handling of ingredients in homes alter the microbial composition of foods? The findings of this study suggest that the microbes we eat as part of normal diets vary in absolute abundance, community composition, and functional potential. This variation depends on the specific ingredients in the meals, whether and how the foods are prepared and processed, and other potential factors, not explored here, including the provenance of ingredients. The significance of this variation on the gut microbiota composition and function, and its impact on human health remains to be elucidated. In addition, much as certain gut microbes can transform and modify dietary constituents and nutrients such as polyphenolic compounds and vitamins in the gut (Tuohy et al., 2012), it is possible that food microbes similarly modify nutritive molecules. Future studies need to explore these questions in rigorous study designs aimed at addressing key questions about the composition and content of food microbes and how these vary across diets and meals, and their impacts on the short term and long term composition and function of the gut microbiota.

Additional Information and Declarations

Competing Interests

Author Contributions

DNA Deposition

Jonathan A. Eisen is an Academic Editor for PeerJ.

Jenna M. Lang performed the experiments, analyzed the data, wrote the paper, prepared figures and/or tables, reviewed drafts of the paper.

Jonathan A. Eisen contributed reagents/materials/analysis tools, reviewed drafts of the paper.

Angela M. Zivkovic conceived and designed the experiments, performed the experiments, analyzed the data, contributed reagents/materials/analysis tools, wrote the paper, prepared figures and/or tables, reviewed drafts of the paper, designed daily meal plans for each diet type.

The following information was supplied regarding the deposition of DNA sequences:

ENA: ERX637113–ERX637127 Figshare: http://dx.doi.org/10.6084/m9.figshare.1246737

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
