# Peer review of "The microbes we eat: abundance and taxonomy of microbes consumed in a day’s worth of meals for three diet types"

_PeerJ, doi:10.7717/peerj.659_

## Round 0.1 · original submission · Major Revisions

· Academic Editor

Major Revisions

This study on the microbes present in food (using both culture-dependent and culture-independent methods) is very interesting. However, the reviewers have raised several questions, which should be addressed in order to improve the quality of this MS.

Reviewer 1 ·

Basic reporting

I think that the Basic Reporting standards are for the most part met. The paper provides sufficient background and is well written overall. In Figure 3, I thought that the FONT of the different diets in the right panel should be larger (AMERICAN, USDA, VEGAN). I had trouble reading them. Also in Figure 4 the hierarchical cluster on the top part of the figure was very light grey and so hard to see.

Also some other minor reporting things:
1) The first line of the abstract is inconsistent between the one in the main paper and the one prefacing the submission. The main paper says “microbes in our poop” and the other says “microbes in our feces.” I prefer “feces.”
2) Figure, Tables are not consistently ordered by their first mention in the text.
3) Page 5: They mention in the methods “We used a custom script available in a GitHub repository” but do not provide a link to the GitHub repository.
4) Middle of page 6 “using the pick_closed_reference_otus.py” should be “using the pick_closed_reference_otus.py script.”

Experimental design

In this paper, Lang et al explore the microbes present in food using both culture-dependent and culture-independent techniques. To do so they prepare or purchase meals, blend them up, and assess the load of microbes as the CFUs and the composition using sequencing of the 16S rRNA gene. They analyze what amounts to a full day of meals (Breakfast, lunch, dinner, and snacks) that are consistent with what may be consumed by someone with 3 different types of diets: 1) The average American diet (convenience foods), 2) The USDA recommended diet or 3) a VEGAN diet. Overall, their results suggest that people ingest millions to billions of bacteria in a typical day and give us some idea of the types of bacteria that these are. I think that the strength of this paper is that it gives more information on what sort of bacteria a person may be introduced to through their diet, which I agree we do not know much about already. A weakness of this paper is its small scale. Although the authors state their main hypothesis in the middle of page 3 to be: “We hypothesized that the microbes that we eat vary both quantitatively and compositionally in a significant way according to dietary pattern.”, they really only look at one to two of each type of meal (breakfast (1), lunch (1), dinner(1), snack (2)) for each type of diet. There are of course many different configurations of foods that could make up a meal that could be considered “AMERICAN”, “USDA recommended”, or “VEGAN” and this paper does not really seem to be powered enough to detect systematic differences in the microbial loads of these different types of diets even if they did exist. Another weakness is the lack of any attempt to see if these microbes in the food survive in the gut or if their DNA can be found there. Overall I would say that this is an “exploratory” paper that provides some preliminary information on an interesting/important topic that we truly know very little about, but is not powered enough to address their main hypothesis and thus provides mostly descriptive information.

Some specific suggestions for improvements below:

1) In the methods I was confused as to how they were able to align their paired end reads. They said that they amplify with 515F and 806R primers, which amplifies a ~300 bp region. Then they say that the sequence with miSeq to generate 150bp amplicon reads. Then they say that they align these reads and compute a consensus and require a 70-120 bp overlap. This does not add up. It would if they were using the 2x250 MiSeq kit. Is this the case?
2) There is some attempt in here to describe the taxa that are driving the PCoA clustering in Figure 1. The description of this in the middle of page 9 was a little confusing, particularly because it said that 7 samples clustered together on the left side of PC1 and 4 samples clustered together on the right side, but I think that this was switched (I see a cluster of 7 on the left). One way to better show this is to use the bi-plots functionality in QIIME described in this url (http://qiime.org/scripts/make_3d_plots.html). It plots the taxa in the same PCoA space as the samples. This is just a suggestion but at the least should correct the left/right PC switching the text).

Validity of the findings

I think that some of their findings are not statistically supported or conclusions do not follow from the data. Specifically:

1) On page 9 the authors describe their attempts to correlate microbes with the nutritional content of food. They say that “due to the exploratory nature of this study, there were no specific hypothesis tested with these correlations and no corrections for multiple hypothesis testing.” I thought that exploratory studies with no specific a-priori hypothesis were precisely when it is important to test for multiple comparisons. For instance, the observation that Blautia was negatively correlated with sugar content of the meal is put right up front in the abstract, but this is not a significant results considering the number of comparisons made. Is the number of “significant” observations (p-value less than .05) more than would be expected just by chance? (i.e. are 5% of the comparisons made significant at this level? If so this is what you would expect to see by chance…)
2) The PiCrust results on page 10 seems to cherry pick two results (the N-glycans and sporulation) that make a good story because they make biological sense (e.g. spores should survive cooking). However the description of the PiCrust results would benefit from a more general description of the totality of the results. Were these the only two significant KEGG functional categories? I think that it is fine to focus on these two in the main text bur should also couple with a more general description of the overall results and reference to a supplementary table showing results of all significant functions. Are these p-values fdr corrected?
3) I think that some of the conclusions are limited given that there is no way to know whether these bacteria can live in the gut. For instance, in multiple places, the authors try to make the point the what we see in the gut can also be a function of what we are eating and these microbes might bring with the diet the appropriate microbes to digest that diet, for instance on page 11 when they say with regard to N-glycan degrading genes in the vegan diet “This suggests that when one consumes a diet that is high in cellulose, one also consumes a population of microbes that is well equipped to digest cellulose.” The problem with this statement is that the population needs to also be well equipped to live in the human gut, pass through the stomach unharmed, etc., and there is really only a small proportion of all of the microbes on the planet who can do that. One way to support some of their statements with this data is to ask what proportion of bacteria that we are seeing in food are microbes that are known to also be active in the gut or are often found in gut surveys and whether these have the N-glycan encoding genes. If they are found in both the gut and the diet however, their presence in surveys of gut populations still does not support entirely that they are active in the gut but would kinda support that they may be?

Reviewer 2 ·

Basic reporting

The authors adhered to all PeerJ policies on basic reporting. Though the recommended sequential line numbering would have been appreciated.

Experimental design

No comment specifically on experimental design. Please see general comments that refer to reporting experimental replicates.

Validity of the findings

No comments.

Additional comments

The authors present a nutritional biology concept that is as provocative as it is under-explored. Dietary microbes not ascribed a ‘probiotic’ function or linked to pathology have been largely ignored. Though these microbes potentially represent inocula, exogenous genetic information, nutritive substrates for host and/or microbiota, and bioactive influences.

Perhaps the greatest advance reported in this manuscript is the systematic quantitative description of sample meals, which is linked to microbiome content. This is reflected in the authors’ summary section that highlights alludes to problematic participant records and recall. Perhaps this point should be stated earlier with greater emphasis to maximize this important scientific motivation.

There are, however, fundamental limitations to this study that should be explicitly stated in the text. This would strengthen the report’s impact in this reviewer’s opinion. For example, it is unclear what replicates were examined per diet pattern. It seems that this study represents a snapshot profile should be remarked upon. As such, comments are absent regarding conservation of microbial community structure in additional sample sources and studies that this article may engender. Would we observe population variance if we swapped foods/sources/preparations/time/geography/climate/etc but kept the overall character of the diet pattern constant (%fat %CHO etc)? I think this would be a likely outcome based on current theory of microbial distribution. Moreover, the methods employed to count viable cells have inherent biases to them. This should be stated in the text. There are a few other (more minor) limitations that come to mind that would better inform the readership. I suspect this study would have broad appeal beyond those working at the host-microbial interface.

Minor concerns:
Abstract- dietary microbes are implicated in disease processes such as infections or intoxications. Perhaps the authors are referring to the potential to act indirectly by enhancing pathogenesis by heterologous microbes?

Introduction- it might be worthwhile to discuss the potential for transformation of nutritive molecules by dietary microbes in addition to host or microbial metabolism?

Introduction- the authors may wish to briefly describe any prevailing uncertainty surrounding the enterotype concept as described in ref 2. This may not be essential, but the statement “diet…plays a key role in enterotype” may not be supported by current evidence. Rather diet influences microbiota composition either in periodic shifts or long-term dietary habits. The authors do describe this well in subsequent sentences.

Intro, pg 2- The human gut microbiome is composed of several ecosystems and not just one depending on how one defines it.

Intro, pg 3- previous antibiotics/microbiome studies- it is appropriate to cite Dethlefsen et al seminal work in this area.

Intro, pg 3- Perhaps some food products/meals/derivatives are not ‘microbial ecologies’. They do not always assemble communities exhibiting species interactions- but rather transient aggregations of microbes and their molecular signatures.

It is unclear what is meant by ‘quantitatively and compositionally’. Aren’t compositional features, presumably community structure, quantitative?

Cleaning products are antimicrobial by nature- e.g. surfactants. Perhaps it would be helpful to clarify if these products contained added antimicrobials.

Fungal vs. yeast plate count should be distinguished. Also, 16s is only referring to bacterial composition in this instance.

It may be worthwhile to remove “Diets were designed by a nutritional biologist...” Is this an author? Is this someone to be acknowledged? In any event, it may be misinterpreted as an appeal to authority instead of what the authors had intended.

Is section 3.1 Meal Composition a best reported in results or methods? I could see the argument for either.

3.6 What is meant by, “important functional roles?” Is this within the community? Also, do we assume that each member plays an “important functional role?”

Figure 3 looks like it is missing a descriptor for the three dietary patterns I assume are depicted.

Sporulation associated KEGG pathway is enriched in the cooked food set. The authors do not comment meaningfully on the potential significance. It is appreciated that any link would be highly speculative, and the authors may have seeking to avoid over-interpretation. Though this passage may benefit from increased context.

---

## Round 0.2 · accepted · Accept

· Academic Editor

Accept

This MS is now accepted. Congratulations!!!